# A Schiff-Base Modified Pt Nano-Catalyst for Highly Efficient Synthesis of Aromatic Azo Compounds

**Yanyan Teng, Xinkui Wang \*, Min Wang, Qinggang Liu, Yuqing Shao, Haomeng Li, Changhai Liang, Xiao Chen and Huilong Wang \***

School of Chemistry, Dalian University of Technology, Dalian 116024, China; 2637259148@mail.dlut.edu.cn (Y.T.); wangmin@dlut.edu.cn (M.W.); liuqinggang@mail.tsinghua.edu.cn (Q.L.); yuqingshao1996@mail.dlut.edu.cn (Y.S.); lihaomeng@mail.dlut.edu.cn(H.L.); changhai@dlut.edu.cn (C.L.); xiaochen@dlut.edu.cn (X.C.)

\* Correspondence: wangxinkui@dlut.edu.cn (X.W.); hlwang@dlut.edu.cn (H.W.);
  Tel.: +86-411-84986073 (X.W.)

**Abstract:** A Schiff-base modified Pt nano-catalyst was prepared via one-pot aldimine condensation and then impregnation-reduction of a platinum precursor, in which the Pt nanoparticles (NPs) with an average size of 2.3 nm were highly dispersed on the support. The as-prepared catalyst exhibited excellent activity and selectivity in the hydrogenation coupling synthesis of aromatic azo compounds from nitroaromatic under mild conditions. The strong metal–support interaction derived from the coordination of nitrogen sites on Schiff-base to Pt NPs enables stabilizing the Pt NPs and achieving the catalytic recyclability. The scheme can also tolerate various functional groups and offer an efficient method for the green synthesis of aromatic azo compounds.

**Keywords:** Pt nano-catalyst; Schiff-base; aromatic azo compounds (Aazo); nitroaromatic hydrogenation

## 1. Introduction

Aromatic azo compounds (Aazo) are important raw materials, which are widely used in the fields of pigments [1–3], radical reaction initiators [4–6], food additives [7,8], electronic devices [9–12], and drugs [13–16]. The industrial production of Aazo is achieved by the azo-coupling reaction between a diazonium salt and an electron-rich arene, in which stoichiometric amounts of nitrite salts are used, therefore generating large amounts of dangerous wastes of diazonium salts in this process [17–19]. Other methods, including the oxidation of anilines by transition metals, the reduction of nitroaromatics by stoichiometric amounts of metals, and the rearrangement of aromatic azoxy compounds, have also been developed for the synthesis of Aazo [20–25]. However, these conventional procedures generally require harsh reaction conditions or a long reaction time. Recently, the catalytic reduction of nitroaromatics to Aazo over noble metal catalysts system has been widely studied. For example, Corma et al. reported the Aazo formation from nitroaromatics via a one-pot, two-step reaction using Au/TiO₂ [26]. Gu et al. reported Pt nanowires and Pd nanoclusters as efficient catalysts could promote the synthesis of Aazo from the corresponding nitroaromatics in the alkaline condition [27–29]. Cao et al. reported the synthesis of Aazo by hydrogenative coupling of nitroarenes over Au/Mg-Al hydrotalcite catalyst without any external additives [30]. Although the catalytic synthesis of Aazo derivatives has achieved high selectivity and good yields, the catalytic efficiency of the noble metal catalysts (the average turnover frequency (TOF) typically less than 300 h⁻¹), at mild reaction conditions, is still needed to improve for the application in industry. Hence, the development of a more efficient heterogeneous catalytic synthetic procedure is highly desirable.

For supported metal catalysts, the support not only disperses the active sites but also modifies the electronic properties of the metals. Furthermore, the support can also provide an active site for

the reactant activation and affect the product desorption. Previous studies have shown that nitro groups could be easily adsorbed on basic or reducible supports [31,32]. It is promising to develop a novel support to tune the catalytic activity in the hydrogenation of nitroaromatics. For example, Pt NPs immobilized on nitrogen-doped carbon nanotubes showed improved catalytic performance in the hydrogenation of nitrobenzene to anilines under mild conditions [33].

Herein, a Schiff-base modified silica-supported Pt catalyst has been prepared for the hydrogenation of nitroaromatics to Aazo. The presence of Schiff-base groups greatly enhances the catalytic activity, selectivity, and stability. The highest yield achieved is ca. 91.1% for 4,4'-dichloroazobenzene (DCAB) over $Pt_{0.5}/SiO_2$-Schiff catalyst with the average TOF up to 2601 $h^{-1}$. The detailed characterization of the structural and electronic properties of the Pt NPs by High-angle annual dark-field scanning transmission electron microscopy (HAADF-STEM), X-ray diffraction (XRD), Fourier transform infrared spectroscopy (FT-IR) and X-ray photoelectron (XPS) spectroscopy reveal "the role of Schiff base".

## 2. Results and Discussion

### 2.1. Catalysts Characterization

XRD patterns of the samples are shown in Figure 1. For the fresh $Pt_{0.5}/SiO_2$ catalyst, besides a broad peak ascribed to the $SiO_2$, there are two diffraction peaks at $2\theta = 39.9°$ and $46.3°$ corresponding to the (1 1 1) and (2 0 0) planes of the face-centered cubic structures of platinum (ICDD PDF No. 64-0802), respectively (Figure 1a) [34]. According to Scherrer's equation, the particle size of Pt NPs is estimated to be ca. 3.8 nm. However, only a very weak and broad peak of Pt at 39.8° is discernible for the fresh $Pt_{0.5}/SiO_2$-Shiff catalyst (Figure 1c), indicating Pt NPs with small particles are highly dispersed on the support. In agreement with the XRD results, the HAADF-STEM image (Figure 2a) shows that the size of Pt NPs on the $Pt_{0.5}/SiO_2$ is ca. 3.2 ± 1.0 nm. However, in the case of $Pt_{0.5}/SiO_2$-Shiff, that is 2.3 ± 0.8 nm (Figure 2c).

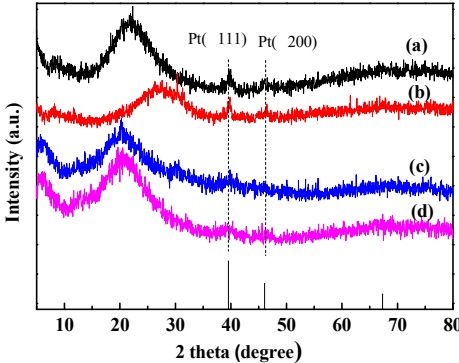

**Figure 1.** XRD patterns of $Pt_{0.5}/SiO_2$ before (**a**) and after reaction (**b**), as well as $Pt_{0.5}/SiO_2$-Shiff before (**c**) and after reaction (**d**).

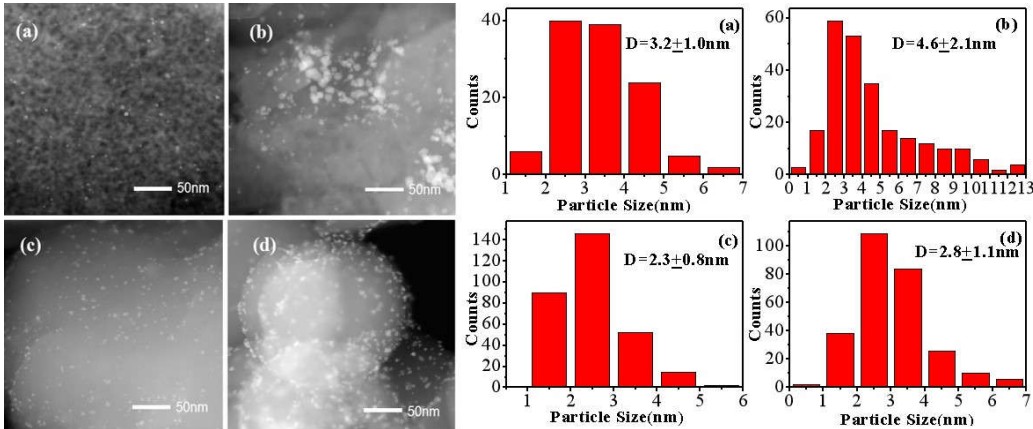

**Figure 2.** HAADF-STEM images and corresponding size distribution of $Pt_{0.5}/SiO_2$ before (**a**) and after reaction (**b**), as well as $Pt_{0.5}/SiO_2$-Shiff before (**c**) and after reaction (**d**).

The FT-IR characterization of the fresh $Pt_{0.5}/SiO_2$ and $Pt_{0.5}/SiO_2$-Shiff has been performed to clarify the surface moieties of the catalysts. As seen in Figure 3, for the $Pt_{0.5}/SiO_2$ catalyst, only characteristic absorption bands for the pure $SiO_2$ are detected, that is, the band at 1107 cm$^{-1}$ is assigned to the antisymmetric stretching vibration of the Si-O. In contrast, for the $Pt_{0.5}/SiO_2$-Shiff catalyst, additional bands at 2869 and 2927 cm$^{-1}$ can be assigned to the symmetric and asymmetric vibrations of the -$CH_2$ groups. The N-H moiety vibration band is observed at 934 cm$^{-1}$, which is attributed to the uncondensed amino group. In addition, the appearance of the Schiff base (-N=C) band at 1695 cm$^{-1}$ indicates the successful graft of Schiff base on the surface of $SiO_2$ support [35].

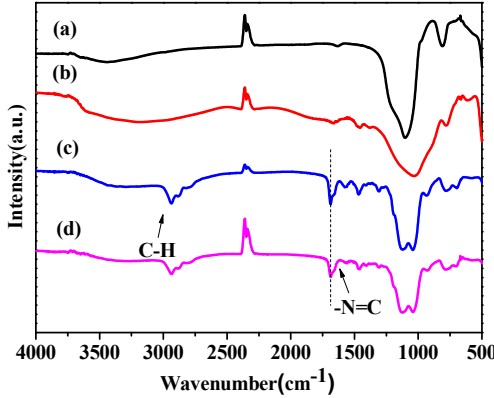

**Figure 3.** FT-IR spectra of $Pt_{0.5}/SiO_2$ before (**a**) and after reaction (**b**), as well as $Pt_{0.5}/SiO_2$-Shiff before (**c**) and after reaction (**d**).

XPS was further performed to characterize the electronic properties of Pt NPs. For the fresh 0.5 wt % $Pt/SiO_2$ (Figure 4a), the binding energy centered at 74.39 and 71.19 eV can be assigned to $Pt4f_{5/2}$ and $Pt4f_{7/2}$ of $Pt^0$, respectively. While for the 0.5 wt % $Pt/SiO_2$-Shiff catalyst (Figure 4b), the binding energy of $Pt4f_{5/2}$ and $Pt4f_{7/2}$ of $Pt^0$ shifted to 73.89 and 70.49 eV. Such a negative shift can be attributed to the electron donation from nitrogen atom to Pt [36,37]. Therefore, there is a strong interaction between the Pt NPs and the organic groups on $SiO_2$-Schiff.

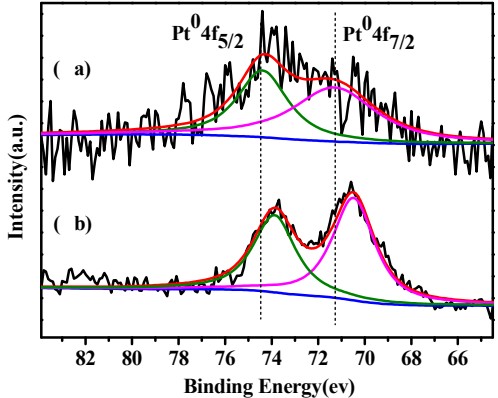

**Figure 4.** Core level XPS spectra in Pt4f regions of $Pt_{0.5}/SiO_2$ (**a**) and $Pt_{0.5}/SiO_2$-Schiff (**b**) catalysts.

## 2.2. Catalytic Activity

The catalytic performance of the as-synthesized Pt nano-catalysts was evaluated in the hydrogenation of *p*-chloronitrobenzene (*p*-CNB), the hydrogenation reaction is shown in Scheme 1. As shown in Table 1, under mild hydrogenation conditions, i.e., 0.5 MPa of $H_2$ pressure in ethanol at 50 °C and KOH as additives, the $Pt_{0.5}/SiO_2$-Schiff shows an excellent hydrogenation activity and azo selectivity, with a yield of 91.1% at the initial 30 min. The average TOF over $Pt_{0.5}/SiO_2$-Schiff catalyst is up to 2601 $h^{-1}$ (Table 1, entry 3). For comparison, the $Pt_{0.5}/SiO_2$ catalyst prefers to the formation of 4,4'-dichloroazoxybenzene (DCAOB) in a yield of ca. 79.0% (Table 1, entry 1).

**Scheme 1.** Hydrogenation of *p*-chloronitrobenzene (*p*-CNB) over different catalysts.

**Table 1.** Hydrogenation of *p*-chloronitrobenzene (*p*-CNB) to 4,4'-dichloroazobenzene (DCAB) over different catalysts.

| Entry | Catalyst | Conv. (%) | Sel. (%) | | |
|---|---|---|---|---|---|
| | | | DCAB | DCAOB | *p*-CNA |
| 1 | $Pt_{0.5}/SiO_2$ | 92.4 | 12.1 | 85.5 | 2.4 |
| 2 [a] | $Pt_{0.5}/SiO_2$ | 10.6 | - | 94.1 | 5.9 |
| 3 | $Pt_{0.5}/SiO_2$-Schiff | 99.0 | 92.1 | 4.7 | 3.2 |
| 4 [a] | $Pt_{0.5}/SiO_2$-Schiff | 98.6 | 89.1 | 5.8 | 5.1 |
| 5 [b] | $Pt_{0.5}/SiO_2$-Schiff | 98.2 | 87.1 | 7.0 | 5.9 |
| 6 | $Pt_{0.3}/SiO_2$-Schiff | 97.2 | 57.0 | 40.1 | 2.9 |
| 7 | $Pt_{0.8}/SiO_2$-Schiff | 100 | 93.0 | 2.1 | 4.9 |

Reaction conditions: *p*-CNB (1 mmol), catalyst (30 mg), 0.5 MPa $H_2$, 2 mmol KOH, ethanol (5 mL), 50 °C, 0.5 h. [a] Catalyst was filtered off after the 1st reaction cycle, then the substrate was added for the second reaction cycle. [b] Catalyst after the 2nd reaction cycle was adopted.

In addition, the $Pt/SiO_2$-Schiff catalyst presents much better stability than that of the $Pt/SiO_2$ catalyst, and only a slight decrease in catalytic efficiency is found in the two successive cycles. However, the catalytic activity of $Pt_{0.5}/SiO_2$ catalyst decreases obviously in cycle experiment (conversion from 92.6 to 10.6%) compared with that of the $Pt_{0.5}/SiO_2$-Schiff catalyst. To understand the unexpected catalytic performance of the $Pt/SiO_2$-Schiff catalyst, we have made a careful comparison of the characterization of the two samples. The inductively coupled plasma-atomic emission spectroscopy (ICP-AES) results show that the Pt loading for the $Pt_{0.5}/SiO_2$ catalyst has a

great decrease from 0.48 to 0.31% after one cycle. However, the metal loading of the $Pt_{0.5}/SiO_2$-Schiff catalyst keeps almost constant after the reaction. The presence of Schiff-base will prevent the leaching of platinum during the reaction process. No obvious change of particle size and peak position are observed in the XRD pattern of the spent $Pt_{0.5}/SiO_2$-Schiff (Figure 1), which indicates that $Pt_{0.5}/SiO_2$-Schiff catalyst has better alkali resistance due to the Schiff base functional group on the surface. While for the used $Pt_{0.5}/SiO_2$ catalyst, the peak position of the silica is slightly shifted because potassium silicate (ICDD PDF No. 48-0866) is formed via the reaction of $SiO_2$ support with KOH. Correspondingly, as shown in the HAADF-STEM images, the stability of $Pt_{0.5}/SiO_2$-Schiff is much better than that of $Pt_{0.5}/SiO_2$ catalyst. The Pt particle size of the spent $Pt_{0.5}/SiO_2$-Schiff has an average size of 2.8 ± 1.1 nm (Figure 2d), while the Pt particle size of the spent $Pt_{0.5}/SiO_2$ is ca. 4.6 nm with some bigger particles even larger than 10 nm (Figure 2b). The enhanced activity and stability of $Pt/SiO_2$-Schiff can be attributed to the strong interaction between the organic groups around and Pt NPs, which keeps the catalytic efficiency of active sites in the reaction.

The effect of Pt loadings in the $Pt/SiO_2$-Schiff catalyst for the hydrogenation of *p*-CNB has also been investigated. The ICP-AES results show that the actual loadings of Pt are 0.76, 0.48 and 0.28 wt % for $Pt_{0.8}/SiO_2$-Shiff, $Pt_{0.5}/SiO_2$-Shiff, and $Pt_{0.3}/SiO_2$-Shiff, respectively. TEM images of $Pt_{0.8}/SiO_2$-Shiff, $Pt_{0.5}/SiO_2$-Shiff, and $Pt_{0.3}/SiO_2$-Shiff show the similar metal particle size, 2.6, 2.2, and 2.4 nm, respectively, as shown in Figure S1. $N_2$-physical adsorption results show that these catalyst samples have the same specific surface area (ca. 18 $m^2 \times g^{-1}$). Under the same reaction conditions, the $Pt_{0.3}/SiO_2$-Schiff catalyst presents a relatively low selectivity due to the lowest Pt content. However, the selectivity to DCAB can be reached to 90% after extending the reaction time to 1 h.

Generally, the catalytic activity and selectivity are sensitive to the base employed in the formation of Aazo [27–29,38–41]. In our results, the amount and strength of the base have also a great influence on the catalytic nitroaromatic hydrogenation. Without the presentation of base, the *p*-chloroaniline (*p*-CNA) is the main product (with a selectivity of 97.3%) at 96.5% conversion of nitrobenzene. Once the base was added in the reaction system, even with a weak base, such as $K_2CO_3$ and $Na_2CO_3$, DCAOB and *p*-CNA would become the main products. As a strong base, such as KOH, was used in the reaction, DCAB was then served as the main product, suggesting that a strong base was indispensable in the course of DCAB formation. The basic additive changes the reaction pathway of the nitroaromatic hydrogenation, in which the dehydration coupling reaction of the aromatic hydroxylamine compound with the aromatic nitroso compound to form an oxygen azo compound is promoted. Subsequently, the oxygen azo compound is further hydrogenated to form an azo compound, enhancing the selectivity to the DCAB. In addition, the amount of KOH significantly affects the product selectivity (Table 1), and the superior yield (91.1%) of DCAB is achieved when 2.0 equivalents of KOH is used, which is the most effective way so far for the azobenzene synthesis.

Solvent effects were also investigated under similar reaction conditions (Table 2). In comparison with other commonly used solvents, our results suggest that ethanol serves as the best solvent for the generation of DCAB (91.1% yield). The reactions in *p*-xylene, toluene, *n*-heptane show a relatively low activity, with DCAOB as a predominant product, which is the intermediate in the DCAB formation. Generally, less polar protic solvents, such as methanol, ethanol, and *i*-propanol, result in the desired product of DCAB. It has been widely reported that the protic solvents could be served as hydrogen donors in hydrogenation reactions. Therefore, the hydrogen transfer process in protic solvents benefits the hydrogenation involved in the DCAB formation reaction.

**Table 2.** Effect of solvent and base on the coupling reactions of *p*-CNB.

| Entry | Solvent | Base | Conv. (%) | Sel. (%) | | |
|:---:|:---:|:---:|:---:|:---:|:---:|:---:|
| | | | | DCAB | DCAOB | *p*-CNA |
| 1 | ethanol | - | 96.5 | 0.3 | 2.4 | 97.3 |
| 2 | ethanol | $K_2CO_3$ (1 eqv.) | 100 | 8.9 | 52.9 | 38.2 |
| 3 | ethanol | $Na_2CO_3$ (1 eqv.) | 98.2 | 6.2 | 27.1 | 66.7 |
| 4 | ethanol | KOH (0.5 eqv.) | 96.5 | 66.2 | 24.4 | 9.4 |
| 5 | ethanol | KOH (1 eqv.) | 97.8 | 78.8 | 10.4 | 10.8 |
| 6 | ethanol | KOH (2 eqv.) | 99.0 | 92.1 | 4.7 | 3.2 |
| 7 | ethanol | KOH (3 eqv.) | 98.9 | 88.1 | 6.8 | 5.1 |
| 8 | methanol | KOH (2 eqv.) | 86.2 | 37.4 | 51.6 | 11.0 |
| 9 | *i*-propanol | KOH (2 eqv.) | 99.7 | 90.9 | 3.4 | 5.7 |
| 10 | toluene | KOH (2 eqv.) | 8.1 | 5.7 | 88.3 | 6.0 |
| 11 | *p*-xylene | KOH (2 eqv.) | 6.2 | 49.5 | 43.2 | 7.3 |
| 12 | *n*-heptane | KOH (2 eqv.) | 14.2 | 12.1 | 43.1 | 44.8 |

Reaction conditions: *p*-CNB (1 mmol), catalyst (30 mg), 0.5 MPa $H_2$, Solvent (5 mL), 50 °C, 0.5 h.

To explore the reaction mechanism, the evolution of product distribution was monitored with the reaction time (Figure 5). As we can see, the DCAOB intermediate was predominant at the initial stage, which is further transformed into DCAB. The other hydrogenative intermediates, such as nitrosobenzene and phenylhydroxylamine, are not detected. Based on the above experimental results, the mechanism of the hydrogenation of *p*-CNB to the corresponding DCAB is proposed in Figure 6, which is consistent with the literature report [31,42]. The Schiff base promotes the adsorption of *p*-CNB, and the Pt NPs are served as active sites to activate $H_2$. The H on the Pt surface reacts with the adsorbed *p*-CNB to generate *p*-CNA, which would be converted to 4-chloro-*N*-hydroxybenzenamine. This 4-chloro-*N*-hydroxybenzenamine can be easily reduced to aniline under a neutral condition, and no coupling product was detected in our experiments. However, when a base is added, the generated 4-chloro-*N*-hydroxybenzenamine will quickly couple with nitrosobenzene to form 4,4′-dichloro-*N*,*N*′-dihydroxy-diphenylhydrazine. This dihydroxy intermediate is then dehydrated to give DCAOB and further hydrogenated to DCAB under strong alkali environment. As a result, DCAB is the only observed coupling product.

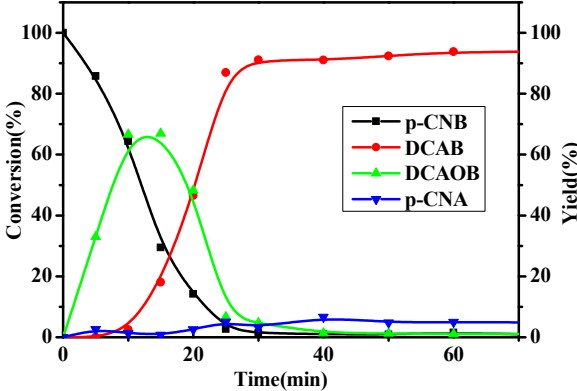

**Figure 5.** Substrate and product concentration profiles as a function of the reaction time.

**Figure 6.** The mechanism of the hydrogenation of *p*-CNB to the corresponding DCAB.

To confirm our proposed mechanism, a serial experiment of the hydrogenation of DCAOB and DCAB was performed. Indeed, hydrogenation of DCAOB to DCAB does not occur under weakly alkaline conditions even for a long period reaction time. With a strong base, DCAOB can be readily converted into DCAB with a high yield (98%) within 30 min. While the generated DCAB could not continue to be hydrogenated to 4,4'-dichlorohydrazobenzene or *p*-CNA under the given condition. As a result, DCAB is the major product in the selective hydrogenation of *p*-CNB reaction under strongly basic condition.

To demonstrate the general applicability of this Pt/SiO$_2$-Schiff catalyst, a series of nitroaromatic compounds in the synthesis of Aazo (Scheme 2) were tested. As shown in Table 3, most of the substituted nitrobenzenes can result in good performance, especially for the compounds with the presence of electron-withdrawing substituents. With an electron donative group such as alkyl (methyl), arylamine (a yield of 49.7%) is the predominant by-product. While the yield of Aazo can up to 86.1% when *i*-propanol is used as a solvent and the reaction temperature is raised to 90 °C. The unsubstituted nitrobenzenes are relatively inert in the Aazo formation at the given conditions (50 °C, 0.5 MPa H$_2$), and the major by-product is azoxybenzene (yield 58.7%). However, it can convert to azobenzene with a yield as high as 84.7% at the higher reaction temperature (90 °C). Moreover, steric properties of the substituent also significantly affect the reaction. Specifically, the yield of *p*-substituted azo arene is higher than that of the corresponding *m*-substituted azo arene (Table 3, entry 5 and 6). Halogenated nitroarenes can be readily converted to corresponding azo compounds without any dehalogenation, while a side reaction is often unavoided in the conventional hydrogenation condition.

**Scheme 2.** Hydrogenation of different substituted nitrobenzenes to form Aazo over Pt catalyst.

**Table 3.** Hydrogenation of different substituted nitrobenzenes over Pt catalyst.

| Entry | R | T/°C | Conv. (%) | Sel. (%) | | |
|---|---|---|---|---|---|---|
| | | | | DCAB | DCAOB | *p*-CNA |
| 1 | H | 50 | 80.4 | 21.4 | 73 | 5.6 |
| 2 | H | 90 | 99.0 | 85.6 | 3.3 | 11.1 |
| 3 | *p*-CH$_3$ | 50 | 95.9 | 48.1 | - | 51.9 |
| 4 [a] | *p*-CH$_3$ | 90 | 99.0 | 87.0 | - | 13.0 |
| 5 | *m*-Cl | 50 | 97.6 | 89.7 | 2.0 | 8.3 |
| 6 | *p*-Br | 50 | 98.8 | 75.6 | 9.1 | 15.3 |

Reaction conditions: substrate (1 mmol), catalyst (30 mg), 0.5 MPa H$_2$, ethanol (5 mL), 0.5 h. [a] *i*-propanol as solvent.

## 3. Materials and Methods

### 3.1. Catalyst Synthesis

SiO$_2$-Schiff: SiO$_2$-Shiff was synthesized according to our previously reported procedure [43], in which the aldimine condensation of (3-aminopropyl)-triethoxysilane (APTES) with formaldehyde was involved. Typically, a HCHO solution (10 mL, 37%) was dripped into an APTES aqueous solution (500 mL, 0.12 mol × L$^{-1}$) while stirring at 40 °C for 1 h. A white precipitate formed, which was filtered and washed with deionized water. And then, the precipitate was aged at 150 °C for 12 h in a Teflon-lined autoclave (Dalian, China). After cooling down to room temperature, a stable Schiff-base functionalized SiO$_2$ support was obtained after by filtering, washing with an excess amount of deionized water, and then drying in vacuum (Shanghai, China) at 120 °C for 12 h.

Pt/SiO$_2$-Shiff: Briefly, 0.5 g SiO$_2$-Shiff was added into an ethanol solution of H$_2$PtCl$_6$ × 4H$_2$O (0.032 mM, 400 mL), and the mixture was magnetically stirred at 80 °C for 30 min. Then, 15 mL of NaBH$_4$ (20 mg) aqueous solution was added to reduce H$_2$PtCl$_6$ to platinum catalyst. After stirring for 1 h, the solid was filtered and washed with an excess amount of deionized water. After drying in vacuum at 120 °C for 12 h, the Pt/SiO$_2$-Shiff catalyst was obtained.

Pt/SiO$_2$: The silica support (with a Brunauer-Emmett-Teller (BET) surface area of 596 m$^2$ × g$^{-1}$) supplied by Qingdao University was calcined in air at 500 °C for 2 h to remove adsorbed water before used. The Pt/SiO$_2$ was prepared by incipient wet technique. Typically, 2mL H$_2$PtCl$_6$ × 4H$_2$O (0.032 mM) aqueous solution was impregnated with 0.5 g SiO$_2$ to generate precursors containing 0.5 wt % Pt. The mixture was magnetically stirred for 2 h under room temperature and then aged for 24 h, followed by drying in an oven at 120 °C for 12 h. The samples were calcined in air at 400 °C for 3 h and reduced in a flowing H$_2$/Ar mixture (50 mL × min$^{-1}$; 20/30 *v/v*) at 300 °C for 3 h. This final product was denoted as Pt/SiO$_2$.

### 3.2. Catalyst Characterization

Pt loadings of as-prepared catalysts were analyzed by inductively coupled plasma-atomic emission spectroscopy (ICP-AES). An X-ray diffraction (XRD) experiment was performed on a PW3040/60 X' Pert PRO (PANalytical, Almelo, Netherland) diffractometer (with CuKa X-ray source operated at 40 kV and 50 mA). High-angle annual dark-field scanning transmission electron microscopy (HAADF-STEM) images of samples were recorded with the FEI Tecnai G2 F30 S-Twin microscope (Thermo Fisher Scientific, Waltham, MA, USA) operated at 300 kV. Before a typical STEM experiment, the sample was dispersed into ethanol after grinding, and the suspension was deposited onto a clean carbon-enhanced copper grid and then dried in air. Fourier transform infrared (FT-IR) spectroscopy with a resolution of 4 cm$^{-1}$ were acquired by Bruker tensor 27 (Billerica, MA, USA) equipped with a DLATGS detector. X-ray photoelectron spectroscopy (XPS) spectra were achieved under a vacuum pressure of 10$^{-9}$ Torr, with a VG ESCALAB 250 (Thermo Fisher Scientific) equipped with a monochromated Al–Ka radiation source (1486.6 eV), and the binding energy was scaled to the C1s (284.6 eV).

### 3.3. Catalyst Evaluation

Typically, a catalyst (30 mg), *p*-CNB (1.0 mmol), potassium hydroxide (KOH) (2.0 mmol, 2.0 equivalence vs. the number of reaction substrate) was added to an ethanol solvent (5 mL). The reaction mixture was put in a 25 mL reactor and charged with 0.5 MPa H$_2$. The reaction was conducted at 50 °C for a certain time. The product yield was quantified by gas chromatography (GC) with FID (Shanghai Tianmei, GC7900 (Shanghai, China) equipped with an HP-5 column (30m×0.32mm×0.25μm)).

For the recyclability test, the spent catalyst was isolated by centrifugation and then thoroughly washed with deionized water, toluene, and ethanol, and dried at 120 °C in vacuum.

## 4. Conclusions

In summary, a facile and highly efficient synthesis process for Aazo formation has been developed using Pt/SiO$_2$-Schiff catalysts. Compared with the traditional Pt/SiO$_2$ nano-catalysts, this Schiff-base modified Pt nano-catalyst showed high selectivity for the Aazo. The interaction between Pt and Schiff base stabilized Pt NPs and tuned the electronic state of Pt, enhancing the catalytic activity and stability. This new route can also be applied for the fabrication of other Schiff based modified metal nanoparticles (such as Pd and Au) by altering the metal precursors, showing promise in catalysis and other fields.

**Supplementary Materials:** The following are available online at www.mdpi.com/xxx/s1, Figure S1: TEM images and corresponding size distribution of Pt$_{0.8}$/SiO$_2$-Shiff(a), Pt$_{0.5}$/SiO$_2$-Shiff (b) and Pt$_{0.3}$/SiO$_2$-Shiff(c).

**Author Contributions:** Y.T. designed the project and performed the catalyst preparation, characterizations and catalytic tests. X.W. and H.W. proposed, planned, designed and supervised the project. M.W., Q. L., Y.S. and H.L. participated in beneficial discussion. C.L. and X.C. helped to polish the manuscript. All authors reviewed and commented on the manuscript.

**Funding:** This work was supported by the National Natural Science Foundation of China (21676145, 21377018, 21872135) and the Fundamental Research Funds for the Central Universities (DUT18LK34).

**Acknowledgments:** We thank anonymous reviewers for helpful suggestions on the manuscript.

**Conflicts of Interest:** The authors declare no conflict of interest.

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
