# Peer review of "A Schiff-Base Modified Pt Nano-Catalyst for Highly Efficient Synthesis of Aromatic Azo Compounds"

_catalysts, doi:10.3390/catal9040339_

Reviewer 1 Report

The authors have satisfactorily addressed the reviewer comments and the resubmitted manuscript is much improved as a result. Particularly, the inclusion of TEM  images, PXRD reference patterns, and additional explanations provided in the results and discussion section have strengthened the manuscript. I recommend publishing the manuscript in it present form.

Reviewer 2 Report

In the revised version of the paper, the authors have exhaustively addressed my previous comments and requests. I can now recommend the paper for publication in Catalysts in it present form.

This manuscript is a resubmission of an earlier submission. The following is a list of the peer review reports and author responses from that submission.

Round  1

Reviewer 1 Report

The paper by Wang and co-workers reports on the preparation, characterization and testing of Schiff-base modified Pt/SiO2 catalysts for hydrogenation coupling synthesis of aromatic azo compounds from nitroaromatic. The Schiff-base modified catalysts are compared on the not-modified counterpart, in term of structural and electronic properties and catalytic performance. They show high selectivity for the aromatic azo compounds and improved stability are observed, rationalized through enhanced Pt NPs/support interaction. The results are quitely interesting and clearly enough presented. After adequate consideration of the points reported here below, in my opinion the paper will be suitable for publication in Catalysts.

1)      About the PXRD patterns in Figure 1: why for pattern (b) (Pt0.5/SiO2 after reaction) the broad peak ascribed to the SiO2 undergoes a ca. 10° shift? Please clarify.

2)      The improved stability is an important point. In this respect, please add also the size distribution analysis from STEM data for Pt0.5/SiO2 after reaction. Only the image is presently reported as Figure 2b.

3)      While the characterization is performed for Pt0.5/SiO2, for testing purposes, the authors considered a series of catalyst with different Pt loading (Table 1), including also Pt0.3/SiO2 and Pt0.8/SiO2. With this respect, a brief discussion on the effect of Pt loading on both physico-chemical properties and catalysis performance is worth to be added. Also, please check entry 6 in Table 1 where both Pt0.3/SiO2 and Pt0.8/SiO2 are reported together.

4)      Please carefully proof-read the paper: several small typos and mistakes are present. Some examples follow, but I recommend a global check and editing:

- SiO2: “2” always subscript

- page 2 line 45: “on the base” -> “on basic”

- page 2 line 61:“In consistent” -> “In agreement” or “In line”

- page 4 line 113: “The Previous investigation” -> “The previous investigation”

- page 5 line 137: “DCAOB” should read “DCAB”

- Table 3: for consistency, please use the same “Azoxy” label as in Table 1 instead of “Azox”

- page 6 lines 164/165: In the sentence: “While the yield of aromatic azo compound could up to 86.1% when i-propanol as a solvent.” a verb is missing, e.g. “when i-propanol is used as a solvent.”

- page 8 line 224: please replace the “.” After “nano-catalysts” with a “,”

Authors response:

Point 1: About the PXRD patterns in Figure 1: why for pattern (b) (Pt0.5/SiO2 after reaction) the broad peak ascribed to the SiO2 undergoes a ca. 10°shift? Please clarify.

Response 1: Thanks for the comment. For Pt0.5/SiO2 after the reaction, the peak shift of silica (ca. 10o) is attributed to the phase formation of potassium silicate (ICDD PDF No.48-0866) via the reaction of SiO2 support with KOH. However, Pt0.5/SiO2-Schiff catalyst possesses a better alkali resistance due to the Schiff base group covered on the surface. The corresponding discussion has been added in the revised manuscript (Lines 118-123).

Figure 1. XRD patterns of Pt0.5/SiO2 before (a) and after reaction (b), as well as Pt0.5/SiO2-Shiff before (c) and after reaction (d).

Point 2: The improved stability is an important point. In this respect, please add also the size distribution analysis from STEM data for Pt0.5/SiO2 after reaction. Only the image is presently reported as Figure 2b.

Response 2: The particle size distribution analysis from STEM image for the spent Pt0.5/SiO2 catalyst has been added in the article (Figure 2).

    Figure 2. HAADF-STEM images and corresponding size distribution of Pt0.5/SiO2 before (a) and after reaction (b),as well as Pt0.5/SiO2-Shiff before (c) and after reaction (d).

As shown in the STEM images, the stability of Pt0.5/SiO2-Schiff  is much better than that of Pt0.5/SiO2 catalyst. The Pt particle size of the spent Pt0.5/SiO2-Schiff has a average size of 2.8±1.1 nm (Figure 2d), while the Pt particle size of the spent Pt0.5/SiO2 is ca. 4.6 nm with some bigger particles even larger than 10 nm (Figure 2b). The enhanced activity and stability of Pt/SiO2-Schiff can be attributed to the strong interaction between the organic groups around and Pt NPs, which keeps the catalytic efficiency of active sites in the reaction. The corresponding discussion can be found in the revised manuscript (Lines 123-129).

Point 3: While the characterization is performed for Pt0.5/SiO2, for testing purposes, the authors considered a series of catalyst with different Pt loading (Table 1), including also Pt0.3/SiO2 and Pt0.8/SiO2. With this respect, a brief discussion on the effect of Pt loading on both physico-chemical properties and catalysis performance is worth to be added. Also, please check entry 6 in Table 1 where both Pt0.3/SiO2 and Pt0.8/SiO2 are reported together.

Response 3: According to your suggestion, The effect of Pt loadings in the Pt/SiO2-Schiff catalyst for the hydrogenation of p-CNB has also been investigated. The ICP-AES results show that the actual loadings of Pt are 0.76wt.%, 0.48 wt.%, and 0.28 wt.% for Pt0.8/SiO2-Shiff, Pt0.5/SiO2-Shiff, and Pt0.3/SiO2-Shiff, respectively. TEM images of Pt0.8/SiO2-Shiff, Pt0.5/SiO2-Shiff and Pt0.3/SiO2-Shiff show the similar metal particle size, 2.6 nm, 2.2 nm and 2.4 nm, respectively, as shown in Figure S1. N2-physical adsorption results show that these catalyst samples have the same specific surface area (ca. 18 m2·g-1). Under the same reaction conditions, the Pt0.3/SiO2-Schiff catalyst present a relative low selectivity due to the lowest Pt content. However, the selectivity to DCAB can be reached to 90% after extending the reaction time to 1 h. The corresponding discussion has been added in the revised manuscript (Lines 130-138).

Figure S1. TEM images and corresponding size distribution of Pt0.8/SiO2-Shiff(a), Pt0.5/SiO2-Shiff (b) and Pt0.3/SiO2-Shiff(c).

Point 4: Please carefully proof-read the paper: several small typos and mistakes are present. Some examples follow, but I recommend a global check and editing:

- SiO2: “2” always subscript

- page 2 line 45: “on the base” -> “on basic”

- page 2 line 61:“In consistent” -> “In agreement” or “In line”

- page 4 line 113: “The Previous investigation” -> “The previous investigation”

- page 5 line 137: “DCAOB” should read “DCAB”

- Table 3: for consistency, please use the same “Azoxy” label as in Table 1 instead of “Azox”

- page 6 lines 164/165: In the sentence: “While the yield of aromatic azo compound could up to 86.1% when i-propanol as a solvent.” a verb is missing, e.g. “when i-propanol is used as a solvent.”

- page 8 line 224: please replace the “.” After “nano-catalysts” with a “,”

Response 4: Thanks for your insight. The grammar and spelling in the revised manuscript have been checked and corrected .

Reviewer 2 Report

The authors detail a method to prepare a Schiff-base modified SiO2 supported Pt catalyst. The catalyst has been thoroughly characterized and its catalytic activity has been investigated. The Schiff-base modified catalyst is superior in terms of metal loading and stability and efficiency. However, there are some issues with the data interpretation and explanation. This article is suitable for publication following minor mandatory revisions as follows:

1.     The XRD patterns indicate ‘polycrystalline Pt’, however, the patterns have to be matched to a reference in a database. The authors must provide reference lines in the XRD along with the JCPDS/ICDD number of the reference pattern to provide more insight on the crystallinity and composition of the Pt/SiO2 catalyst.

2.     The authors must provide a hypothesized structure for the Schiff base modified Pt/SiO2 catalyst- either through a cartoon or ChemDraw structure. Without this, it is impossible to interpret the FTIR and XPS data of the catalyst.

3.     In the FTIR data, the authors state that N-H vibration is observed. This is not consistent with the structure of a Schiff base and further necessitates the need for a structural representation as mentioned in comment 2. Authors should clarify if the N-H band is from the catalyst or if the peak corresponds to another bond.

4.     Considering the small size of the nanoparticles, TEM would be a more useful characterization tool than SEM to look at the morphology of the catalyst.

5.     What is the reason for using KOH or weaker bases in enhancing the selectivity of the reaction? (Lines 114-119)

6.  Subheadings should be added to the results and discussion section to make the article easier to read. For example, categorizing the various reactions that this catalyst was used on would greatly enhance readability. 

Minor errors:

1.     Table 1 has errors in Entry 6 with respect to the catalyst composition.

2.     Line 113, P is capitalized in previous

3.   Some grammatical errors exist in the manuscript. The authors must proof-read their revised manuscript before resubmission.

Authors response: 

Point 1: The XRD patterns indicate polycrystalline Pt, however, the patterns have to be matched to a reference in a database. The authors must provide reference lines in the XRD along with the JCPDS/ICDD number of the reference pattern to provide more insight on the crystallinity and composition of the Pt/SiO2 catalyst.

Response 1: Thanks for the comment. It is hard to distinguish the Pt is polycrystalline or single crystalline. We tried to use the SAED to detect the crystal property of the Pt. Because the size of Pt is too small, the diffraction pattern of Pt did not appear in the SAED pattern. Pt nanoparticle is more suitable to describe the Pt. So, we revised the description as “Pt nanoparticle” instead of “Pt polycrystalline”. The reference lines in the XRD along with the JCPDS/ICDD number of the reference pattern has also been added to the revised manuscript (Lines 60-63). 

Figure 1.1.  TEM images and SAED patterns of Pt0.5/SiO2  (a) and Pt0.5/SiO2-Shiff (b).

Figure 1. XRD patterns of Pt0.5/SiO2 before (a) and after reaction (b), as well as Pt0.5/SiO2-Shiff before (c) and after reaction (d).

Point 2: The authors must provide a hypothesized structure for the Schiff base modified Pt/SiO2 catalyst- either through a cartoon or ChemDraw structure. Without this, it is impossible to interpret the FT-IR and XPS data of the catalyst.

Response 2: Thanks for your valuable and constructive comment. We have provided a hypothetical structure for Pt/SiO2-Schiff catalyst in the ChemDraw structure as a graphical abstract in the article.

Figure 3. FT-IR of Pt0.5/SiO2 before (a) and after reaction (b), as well as Pt0.5/SiO2-Shiff before (c) and after reaction (d).

Point 3: In the FTIR data, the authors state that N-H vibration is observed. This is not consistent with the structure of a Schiff base and further necessitates the need for a structural representation as mentioned in comment 2. Authors should clarify if the N-H band is from the catalyst or if the peak corresponds to another bond.

Response 3:  SiO2-Shiff was synthesized by our previously reported method, which involved aldimine condensation of (3-aminopropyl) triethoxysilane (APTES) with formaldehyde. A small amount of the amine group of APTES does not undergo an aminaldehyde condensation reaction with formaldehyde. N-H moiety vibration band is observed at 934 cm-1, which is attributed to the uncondensed amine group. The corresponding discussion can be found in the revised manuscript (Lines 77-78). Thanks for your insight.

Point 4: Considering the small size of the nanoparticles, TEM would be a more useful characterization tool than SEM to look at the morphology of the catalyst.

Response 4: We have performed TEM characterization of the catalyst. The Pt NPs are not easy to distinguish in the Pt0.5/SiO2 catalyst, so that it is difficult to calculate the particle size distribution. Compared with the TEM images, STEM is a good tool to detect the morphology of as-prepared catalyst. Therefore, we still retain the STEM images and particle size distribution in the revised manuscript.

Figure 4.1. TEM images of Pt0.5/SiO2 before (a) and after reaction (b), as well as Pt0.5/SiO2-Shiff before (c) and after reaction (d)

Point 5: What is the reason for using KOH or weaker bases in enhancing the selectivity of the reaction? (Lines 114-119)

Response 5:  The mechanism of the base effect has been well studied by references [27–29,38-41]. Generally, The basic additive changes the reaction pathway of the nitroaromatic hydrogenation, in which the dehydration coupling reaction of the aromatic hydroxylamine compound with the aromatic nitroso compound to form an oxygen azo compound is promoted. Subsequently, the oxygen azo compound is further hydrogenated to form an azo compound, enhancing the selectivity to the DCAB. The corresponding discussion has been added in the revised manuscript (Lines 141-155).

27.   Wang, J.; Hu, L.; Cao, X.; Lu, J.; Li, X.; Gu, H. Catalysis by Pd nanoclusters generated in situ of high-efficiency synthesis of aromatic azo compounds from nitroaromatics under H2 atmosphere. RSC Adv. 2013, 3, 4899-4902. doi:10.1039/c3ra23004j.

28.  Hu, L.; Cao, X.; Shi, L.; Qi, F.; Guo, Z.; Lu, J.; Gu, H. A Highly Active Nano-Palladium Catalyst for the Preparation of Aromatic Azos under Mild Conditions. Org. Lett. 2011, 13, 5640–5643. doi:10.1021/ol202362f.

29.  Hu, L.; Cao, X,; Chen, L.; Zheng, J.; Lu, J.; Sun, X.; Gu, H. Highly efficient synthesis of aromatic azos catalyzed by unsupported ultra-thin Pt nanowires. Chem. Commun. 2012, 48, 3445-3447. doi:10.1039/c2cc30281k.

38.  Song, J.; Huang, Z. F.; Pan, L.; Li, K.; Zhang, X.; Wang, L.; Zou, J. J. Review on selective hydrogenation of nitroarene by catalytic, photocatalytic and electrocatalytic reactions. Applied Catalysis B: Environmental 2018, 227, 386–408. doi:10.1016/j.apcatb.2018.01.052

39.  Combita, D.; Concepción, P.; Corma, A. Gold catalysts for the synthesis of aromatic azocompounds from nitroaromatics in one step. Journal of Catalysis 2014 , 311, 339–349. doi:10.1016/j.jcat.2013.12.014

41.  Guo, X.; Hao, C.; Jin, G., Zhu, H. Y.; Guo, X. Y. Copper Nanoparticles on Graphene Support: An Efficient Photocatalyst for Coupling of Nitroaromatics in Visible Light. Angewandte Chemie International Edition 2014, 53, 1973–1977. doi: 10.1002/anie.201309482

Point 6: Subheadings should be added to the results and discussion section to make the article easier to read. For example, categorizing the various reactions that this catalyst was used on would greatly enhance readability. 

Response 6: We appreciate your suggestion in the reviewing process. Subheadings have been added to the article.

Minor errors:

1.     Table 1 has errors in Entry 6 with respect to the catalyst composition.

2.     Line 113, P is capitalized in previous

3.   Some grammatical errors exist in the manuscript. The authors must proof-read their revised manuscript before resubmission.

Thanks for your insight. The grammar and spelling in the article have been checked and corrected.